# Microbiological diagnosis of pleural infections: a comparative evaluation of a novel syndromic real-time PCR panel

Øyvind Kommedal,[1] Tomas Mikal Eagan,[2,3] Øystein Fløtten,[2,3] Truls Michael Leegaard,[4,5] William Siljan,[6] Hilde Fardal,[7] Bjørnar Bø,[8] Fredrik Grøvan,[9] Kjersti Wik Larssen,[10] Arne Kildahl-Andersen,[11] Reidar Hjetland,[12] Rune Tilseth,[13] Sølvi Kristine Øyen Hareide,[1] Marit Tellevik,[1] Ruben Dyrhovden[1]

**ABSTRACT**    Current microbial diagnostics for pleural infections are insufficient. Studies using 16S targeted next-generation sequencing report that only 10%–16% of bacteria present are cultured and that 50%–78% of pleural fluids containing relevant microbial DNA remain culture negative. As a rapid diagnostic alternative suitable for clinical laboratories, we wanted to explore a PCR-based approach. Based on the identification of key pathogens, we developed a syndromic PCR panel for community-acquired pleural infections (CAPIs). This was a pragmatic PCR panel, meaning that it was not designed for detecting all possibly involved bacterial species but for confirming the diagnosis of CAPI, and for detecting bacteria that might influence choice of antimicrobial treatment. We evaluated the PCR panel on 109 confirmed CAPIs previously characterized using culture and 16S targeted next-generation sequencing. The PCR secured the diagnosis of CAPI in 107/109 (98.2%) and detected all present pathogens in 69/109 (63.3%). Culture secured the diagnosis in 54/109 (49.5%) and detected all pathogens in 31/109 (28.4%). Corresponding results for 16S targeted next-generation sequencing were 109/109 (100%) and 98/109 (89.9%). For bacterial species included in the PCR panel, PCR had a sensitivity of 99.5% (184/185), culture of 21.6% (40/185), and 16S targeted next-generation sequencing of 92.4% (171/185). None of the bacterial species present not covered by the PCR panel were judged to impact antimicrobial therapy. A syndromic PCR panel represents a rapid and sensitive alternative to current diagnostic approaches for the microbiological diagnosis of CAPI.

**IMPORTANCE**    Pleural empyema is a severe infection with high mortality and increasing incidence. Long hospital admissions and long courses of antimicrobial treatment drive healthcare and ecological costs. Current methods for microbiological diagnostics of pleural infections are inadequate. Recent studies using 16S targeted next-generation sequencing as a reference standard find culture to recover only 10%–16% of bacteria present and that 50%–78% of samples containing relevant bacterial DNA remain culture negative. To confirm the diagnosis of pleural infection and define optimal antimicrobial therapy while limiting unnecessary use of broad-spectrum antibiotics, there is a need for rapid and sensitive diagnostic approaches. PCR is a rapid method well suited for clinical laboratories. In this paper we show that a novel syndromic PCR panel can secure the diagnosis of pleural infection and detect all bacteria relevant for choice of antimicrobial treatment with a high sensitivity.

**KEYWORDS**    pleural infection, pleural empyema, syndromic PCR, diagnostics, *Fusobacterium nucleatum*, *Streptococcus intermedius*

Address correspondence to Øyvind Kommedal, oyvind.kommedal@helse-bergen.no.

Ø.K. is a co-founder and shareholder in Pathogenomix Inc. T.M.E. has received grants from GlaxoSmithKline and payment/honoraria from Boehringer Ingelheim, AstraZeneca, and SOS International. T.M.E. also reports participation on a data safety monitoring board (DSMB) for a nutrition study, Helse Sør-Øst, Lillehammer. T.M.E. is leader of the committee for the education of specialists in pulmonary medicine, Norway. T.M.L. is a member of the Professional Advisory Board, GenMark Diagnostics (ePlex), and has received honoraria for time spent. T.M.L. is the president of European Union of Medical Specialists (UEMS) Section of Medical Microbiology and member of Professional Affairs Subcommittee, European Society of Clinical Microbiology and Infectious Diseases (ESCMID). All other authors declare no competing interests.

Pleural empyemas represent a diverse group of mono- and polymicrobial infections with high morbidity and a reported in-hospital mortality rate of approximately 10% (1). According to recent reports, the incidence of pleural infections is increasing (2, 3).

Current approaches for microbiological diagnostics of pleural infections are insufficient. Three investigations using 16S ribosomal RNA (rRNA) gene targeted next-generation sequencing (16S TNGS) as the gold standard found culture to recover only 10%–16% of bacteria present and that 50%–78% of the samples positive by 16S TNGS remained culture negative (4–6). Amplification of the bacterial 16S rRNA gene directly from samples followed by Sanger sequencing (direct 16S Sanger sequencing) is a culture-independent method available in many hospital laboratories. Unfortunately, turnaround times are typically 3 or more working days and it has a relatively low sensitivity (7, 8). The usefulness is further limited due to the frequent polymicrobial nature of pleural infections. In one study, it identified 22.5% of bacteria present (4).

To confirm the diagnosis of pleural infection and define optimal antimicrobial therapy while limiting unnecessary use of broad-spectrum antibiotics, there is a need for rapid and sensitive diagnostic approaches. Inspired by recent introductions of commercial syndromic PCR panels (9–12), we wanted to explore the possibility for a syndromic PCR panel for community-acquired pleural infections (CAPIs). For a syndromic PCR panel to be useful, the targeted infection must be caused by a predictable and relatively limited number of microbes. Some pleural infections occur as a complication to community-acquired pneumonia (CAP), and a small group of pathogens causes the majority of these mainly monobacterial infections. However, most CAPIs are caused by facultative and anaerobic bacteria from the human oral microbiota not normally associated with CAP (4–6). Such oral-type pleural infections (OPIs) can be very complex, making it impossible to design a PCR panel that covers all potentially involved species (13). We therefore suggest a pragmatic approach based on certain microbial patterns that appear to be consistent across all empyema in this group (4, 6). Since bacteria involved in OPI have relatively unproblematic and predictable antimicrobial susceptibility profiles, these patterns allow us to both confirm the diagnosis of OPI and guide antimicrobial treatment based on the detection of a limited group of bacteria. A few key pathogens, *Streptococcus intermedius*, *Fusobacterium nucleatum,* and *Aggregatibacter aphrophilus*, are necessary to establish the infection, and one or more of these are always present (4, 6). Consequently, detection for these species should be sufficient to secure the diagnosis of OPI. For development into a polymicrobial infection, *F. nucleatum* or *A. aphrophilus* seems to be essential (4, 6). Such polymicrobial infections are predominantly anaerobic and detection for these two species can be used to assess the need for specific anaerobic antimicrobial coverage (6). Empyema with *S. intermedius* without *A. aphrophilus* and/or *F. nucleatum* can be assumed monomicrobial (6).

The aim of the present study was to design and evaluate a pragmatic syndromic PCR panel for CAPIs. The diagnostic performance of the CAPI-PCR was compared to routine microbial culture and 16S TNGS.

## MATERIALS AND METHODS

The study design is presented in Fig. 1. As a basis for the PCR panel design and for the initial evaluation, we used 36 community-acquired empyema from a previous retrospective study (4). Thereafter, we conducted a prospective multicenter study specifically on CAPIs to acquire more samples for the evaluation of the PCR panel and to provide more robust data on microbial patterns in OPI (6). The prospective study included 77 patients whereof 73 had residual sample material for the current PCR evaluation. All 109 (36 + 73) samples were characterized using culture and 16S TNGS as part of previous publications (4, 6). As a negative patient control group, we included 11 pleural fluid samples from patients with non-infectious conditions that were negative by both culture and 16S TNGS.

In addition, to provide an indication of the diagnostic sensitivity of the CAPI-PCR versus that of culture and 16S TNGS in a clinical setting, we included data from the first

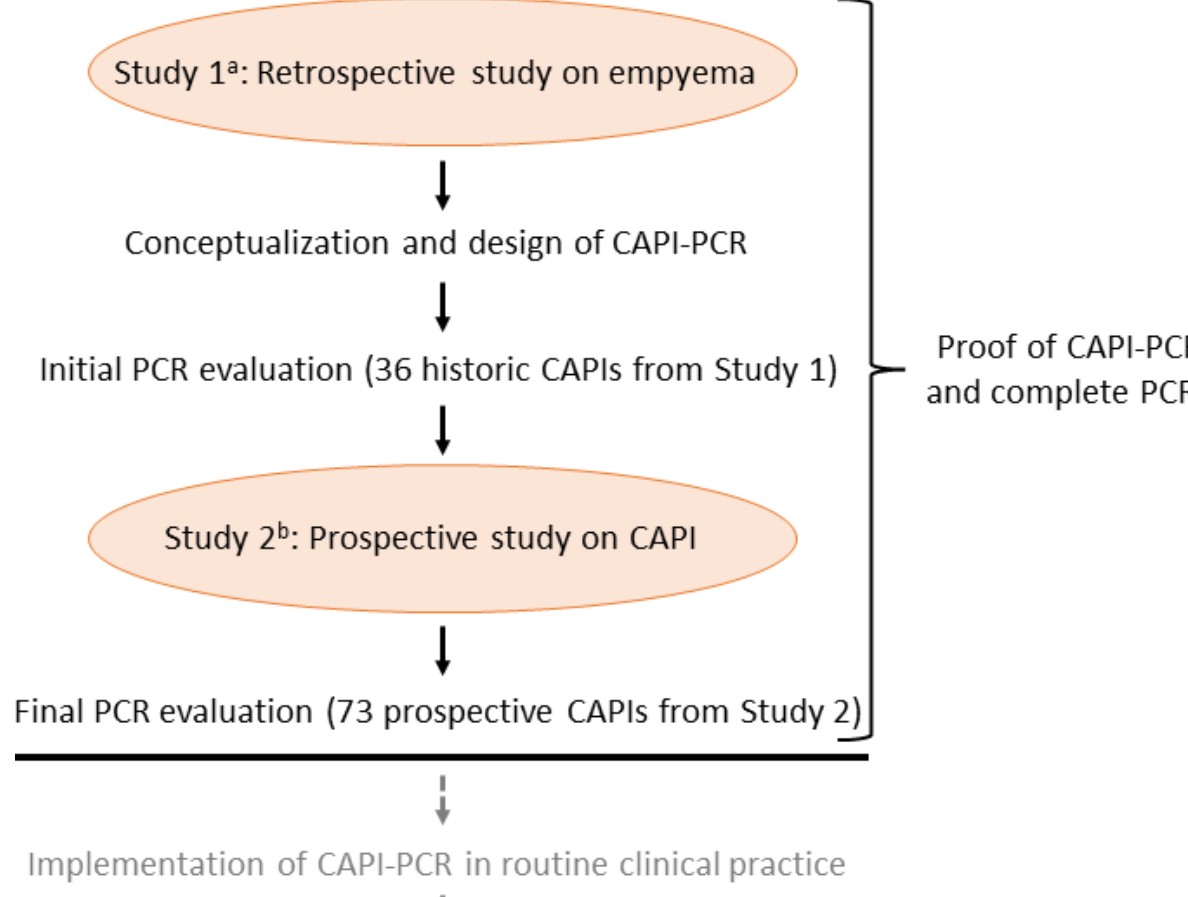

**FIG 1** Study design. a, Dyrhovden et al. (4); b, Dyrhovden et al. (6).

100 pleural fluid samples analyzed with the fully validated CAPI-PCR after implementation in the routine laboratory at Haukeland University Hospital (HUH).

## PCR design and target considerations

We designed real-time PCRs targeting the most common causes of post-pneumonia empyema (*Haemophilus influenzae*, *Pseudomonas aeruginosa*, *Staphylococcus aureus*, *Streptococcus pneumoniae*, *Streptococcus pyogenes,* and selected genera within the *Enterobacterales* order) and the OPI key pathogens (*A. aphrophilus, F. nucleatum,* and *S. intermedius*). These are the two largest groups of empyema overall and represent almost all CAPIs. We also included a PCR for *Parvimonas micra* as an additional indicator of an anaerobic component in OPI.

The PCRs for *H. influenzae*, *S. aureus*, *S. pneumoniae,* and *S. pyogenes* were modifications of previously published assays (14–17). The remaining PCRs were designed for this study. The PCRs were combined into four duplex and one triplex PCRs (multiplex PCRs 1–5, Table 1). For clinical use, we also included an inhibition control based on spiking and detection of the MS2 phage in a separate PCR. To allow for a single setup, all PCRs were designed to fit the same thermal profile. Further details on PCR design and target considerations are provided in Supplementary document S1.

**TABLE 1** Description of the PCRs included in the CAPI-PCR panel[f]

| MP | PCR[a] | Gene[b] | P/Pb | Cons. (µM) | Sequence (5´–3´) | Amp. (bp) | Fl/Q |
|---|---|---|---|---|---|---|---|
| MP1 | Fnecgon[c] | nusG | F | 0.4 | GACCCTACTCCRACAAATC | 93 | FAM[d]/ |
| | | | R | 0.4 | AAGCGAYGAAGGAATYAACTATA | | BHQ1 |
| | | | Pb | 0.2 | CGGACTACATACCAMGCATCKGAATC | | |
| | Fnucl[c] | rpoB | F | 0.4 | CATCACTTACTATGCCWCATG | 237 | |
| | | | R | 0.4 | CTAAGTGWGTTCCATCTKCTAAG | | |
| | | | Pb | 0.2 | TCTGCWGGTAATACTCTTGAAACAACCC | | |
| | Pmicr[c] | rpoB | F | 0.4 | GACGGAGCAAGTGATATTG | 122 | LC610/ |
| | | | R | 0.4 | CCAACAGTTACAGGATTGTC | | BHQ2 |
| | | | Pb | 0.2 | TCATCTCCAGTTCTTCCGTCTCTAAGT | | |
| MP2 | Spyog[e] | tetR | F | 0.4 | TCGCTACTATTTCTTACCTCAA | 94 | FAM/ |
| | | | R | 0.4 | GTCACAATGTCTTGGAAACC | | BHQ1 |
| | | | Pb | 0.2 | CGCAACTCATCAAGGATTTCTGTTACC | | |
| | Sintcon[c] | Cpn60 | F | 0.4 | GTTCCRGTTTCTAATAAAGAAG | 151 | LC610/ |
| | | | R | 0.4 | GCTCTGTKTCCATTCCTT | | BHQ2 |
| | | | Pb | 0.2 | TGATGACACCGTCGTTGCCA | | |
| MP3 | Spneu[g] | lytA | F | 0.4 | CGCAATCTAGCAGATGAAG | 72 | FAM/ |
| | | | R | 0.4 | GTGCGTTTTAATTCCAGCTA | | BHQ1 |
| | | | Pb | 0.2 | CCCTGTATCAAGCGTTTTCGGCAA | | |
| | Hinfl[h] | siaT | F | 0.4 | GGAACTAATGGCCCAATA | 74 | LC610/ |
| | | | R | 0.4 | CGTGATGCTGGTTATGAC | | BHQ2 |
| | | | Pb | 0.2 | AAGCAGCAGTAATTCCTCCGCAA | | |
| MP4 | Aaphr[c] | nusG | F | 0.4 | TGGGCTTTATTGGTGGTA | 143 | FAM/ |
| | | | R | 0.4 | GKTTACGCGCACTTCTTC | | BHQ1 |
| | | | Pb | 0.2 | CGCCAATTAGTAAYCGTGAAGCAGAT | | |
| | Saure[i] | nuc | F | 0.4 | GCCACGTCCATATTTATCAG | 130 | LC610/ |
| | | | R | 0.4 | GCATCCTAAAAAAGGTGTAGAGA | | BHQ2 |
| | | | Pb | 0.2 | TCGTAAATGCACTTGCTTCAGGRCC | | |
| MP5 | Ebact[c] | rpoB | F | 0.6 | CCTGTCTGCTATYGAAGAA | 148 | FAM/ |
| | | | R | 0.4 | GTCCATGTAGTCAACCTG | | BHQ1 |
| | | | Pb | 0.4 | CTGAACARGCTKGATTCGCCTT | | |
| | Paeru[c] | tyrZ | F | 0.4 | CAGGTGATCCTGACCATG | 91 | LC610/ |
| | | | R | 0.4 | CTTCCTGGATACCAATATAGTTG | | BHQ2 |
| | | | Pb | 0.2 | TCTTCTTCACGCCATCCAGCC | | |
| Ctrl | MS2 | | F | 0.4 | TGCTCGCGGATACCCG | 61 | FAM/ |
| | | | R | 0.4 | AACTTGCGTTCTCGAGCGAT | | BHQ1 |
| | | | Pb | 0.2 | ACCTCGGGTTTCCGTCTTGCTCGT | | |

[a]Fnecgon-PCR targets *Fusobacterium necrophorum* and *Fusobacterium gonidiaformans*. Fnucl-PCR targets the *F. nucleatum* group. Pmicr-PCR targets *P. micra* and "*Parvimonas sp. HMT-110*". Spyog-PCR targets *S. pyogenes*. Sintcon-PCR targets *S. intermedius* and *Streptococcus constellatus*. Aaphr-PCR targets *A. aphrophilus* and *Aggregatibacter kilianii*. Saure-PCR targets *S. aureus*. Paeru-PCR targets *P. aeruginosa*. Ebact-PCR targets the genera *Citrobacter*, *Enterobacter*, *Escherichia*, *Hafnia*, *Klebsiella*, *Raoultella*, *Salmonella*, and *Serratia*.

[b]*nusG*, transcription regulation; *rpoB*, RNA polymerase beta subunit; *tetR*, transcriptional regulator; *cpn60*, chaperonin; *lytA*, pneumococcal autolysin; *siaT*, sialic acid transporter permease; *nuc*, thermostable nuclease; *tyrZ*, tyrosyl-tRNA synthetase 2.

[c]This study.

[d]Optionally two separate fluorophores can be used to distinguish between Fnecgon and Fnucl.

[e]Modified from Kodani et al. (14).

[f]MP, multiplex PCR; P/Pb, Primer/Probe; Cons (µM), Final concentration in PCR-reaction tube; Amp. (bp), amplicon size (basepairs); Fl/Q, Fluorophore/Quencher; Ctrl, extraction and inhibition control.

[g]Modified from Carvalho et al. (17).

[h]Modified from Price et al. (15).

[i]Modified from Pichon et al. (16).

## DNA extraction

DNA extraction was performed as described previously (7). Briefly, bacterial cells were mechanically disrupted using a MagNA Lyser apparatus (Roche, Mannheim, Germany), followed by DNA extraction and purification on a MagNA Pure 96 instrument (Roche). After the initial validation, in relation to implementation in the diagnostic routine,

we included an MS2 extraction and inhibition control. After mechanical lysis, prior to the MagNA Pure 96 extraction, all samples were spiked with an MS2 DNA plasmid (TIB Molbiol/Roche, Berlin, Germany). The MS2 concentration was adjusted to become positive around Ct 28 in the MS2 PCR.

## PCR conditions

Primer and probe sequences, together with their respective concentrations in the PCR reaction mixes, are provided in Table 1. Each multiplex PCR was run in a 25 µL reaction volume consisting of 12.5 µL TaKaRa Premix ExTaq (TaKaRa, Kusatsu, Japan), primer and probe concentrations according to Table 1, 1 µL–3.5 µL of PCR-grade water depending on primer/probe volumes, and 2 µL sample template. The PCRs were run in a 96-well plate on a LightCycler 480 II real-time instrument (Roche). The two-step thermal profile included an initial enzyme activation step (95°C/30 seconds) followed by 45 cycles of melting (95°C/10 seconds) and annealing/extension (58°C/30 seconds).

## Interpretation of results

A positive PCR reaction was defined as reaching the fluorescence threshold value (CT) before cycle 40, except for Multiplex-PCR 5 (*P. aeruginosa* and *Enterobacterales*) where we used a cutoff at 33 cycles due to well-known problems with low-level background contamination with DNA from these bacteria in PCR reagents/disposables. Any amplification curves after these cutoffs were reported as negative.

## Orthogonal confirmatory analysis

Positive PCR results that were not confirmed by culture or 16S TNGS underwent orthogonal confirmatory analysis, including re-analysis of 16S TNGS results using a less strict cutoff for removal of low-abundant reads and, for *S. aureus* and *P. micra*, reassessment using alternative PCR assays (Table S1).

## Data management and statistical analysis

For statistical and comparative analysis of the performance of the syndromic PCR, results from 16S TNGS, culture, and orthogonal confirmation analysis were used as a composite reference standard. A finding by the syndromic PCR confirmed by one or more of these analyses was considered a true positive. The sensitivity and specificity for each of the PCRs included in the syndromic PCR panel were calculated individually. In addition, we calculated the overall sensitivity and specificity for the PCR panel for confirmation of bacterial presence in pleural fluid and for the detection of bacteria targeted by the panel.

Sensitivity and specificity values including exact binomial confidence intervals were calculated using R the programming language (Team 2022), version 2022.12.0+35 and R package "excactci" (https://CRAN.R-project.org/package=exactci), version 1.4-4.

## Reporting in routine diagnostics

For the doctors in charge of the patients to have a clear understanding of the clinical meaning of results obtained by the CAPI-PCR, we implemented standard comments as described in Table 2.

## Routine culture and identification of bacterial isolates

Routine samples were cultured aerobically on blood agar and chocolate agar plates and anaerobically on fastidious anaerobic agar (FAA) plates with and without kanamycin and vancomycin. Samples were also cultured in an enrichment broth [brain heart infusion (BHI)]. Blood agars, chocolate agars, and BHIs were incubated in a $CO_2$-enriched atmosphere for 48 hours. FAA plates were incubated in an anaerobe atmosphere for 48 hours. Bacterial colonies were identified using MALDI-TOF MS Bruker Microflex (Bruker Biotyper, Bremen, Germany).

**TABLE 2** Expected result patterns and suggested standard comments/interpretation for the CAPI-PCR[e]

| PCR | Negative/unusual pleural infections | Oral-type pleural infections | | | Non-oral-type pleural infections | | | | | |
|---|---|---|---|---|---|---|---|---|---|---|
| Sintcon | Neg | Pos/Neg | Pos/Neg | Pos[d] | Neg | Neg | Neg | Neg | Neg | Neg |
| Fnucl/Fnecgon | Neg | Pos[c] | Pos[c]/Neg | Neg | Neg | Neg | Neg | Neg | Neg | Neg |
| Aaphr | Neg | Pos[c]/Neg | Pos[c] | Neg | Neg | Neg | Neg | Neg | Neg | Neg |
| Pmicr | Neg | Pos/Neg | Pos/Neg | Neg | Neg | Neg | Neg | Neg | Neg | Neg |
| Spyog | Neg | Neg | Neg | Neg | Pos | Neg | Neg | Neg | Neg | Neg |
| Spneu | Neg | Neg | Neg | Neg | Neg | Pos | Neg | Neg | Neg | Neg |
| Hinfl | Neg | Neg | Neg | Neg | Neg | Neg | Pos | Neg | Neg | Neg |
| Saure | Neg | Neg | Neg | Neg | Neg | Neg | Neg | Pos | Neg | Neg |
| Paeru | Neg | Neg | Neg | Neg | Neg | Neg | Neg | Neg | Pos | Neg |
| Ebact | Neg | Neg | Neg | Neg | Neg | Neg | Neg | Neg | Neg | Pos |
| CAPI-PCR | Neg[a] | Pos[b] | Pos[b] | Pos[b] | Pos[b] | Pos[b] | Pos[b] | Pos[b] | Pos[b] | Pos[b] |

[a]Comment for negative CAPI-PCR: "The sample has been investigated using a PCR targeting the most important bacterial causes of community-acquired empyema in adults. We did not detect any of these bacteria. A negative PCR does not exclude the presence of other bacteria. The sample will also be cultured and analyzed with direct 16S rRNA sequencing."
[b]Comment for all positive CAPI-PCRs: "The microbe(s) has been identified using a PCR targeting the most important bacterial causes of community-aquired empyema in adults. The sample will also be cultured."
[c]Additional comment for positive *F. nucleatum* and/or *A. aphrophilus*: "*F. nucleatum*/*A. aphrophilus* indicates a potential poly-microbial infection. Antimicrobial treatment should include specific anaerobic coverage (e.g., metronidazole) in addition to coverage of facultative oral bacteria (e.g., Penicillin-G or another beta-lactam). Piperacillin-tazobactam also represents a good alternative."
[d]Comment for positive *S. intermedius* only: "Samples that contain *S. intermedius* without concomitant detection of *F. nucleatum* or *A. aphrophilus*, will normally be mono-microbial."
[e]Pos, positive/detected; Neg, negative/not detected.

## RESULTS

### Technical performance of the CAPI-PCR

Technical performance data for the PCRs are provided in Table S2. The expected analytical sensitivity for all PCRs was in the range 1–10 copies per reaction, but detection limits for *P. aeruginosa* and *Enterobacterales* were increased about 10-fold due to the applied cutoff of 33 cycles for these two PCRs. During this evaluation, we observed only a single false positive reaction at CT 34.7 for the *Enterobacterales*-PCR and none for *Pseudomonas*. However, such background contamination can vary over time, between different batches of reagents and between different vendors of consumables/reagents.

### Diagnostic performance of the CAPI-PCR

The diagnostic performance of the CAPI-PCR, culture, and 16S TNGS as compared to the composite reference standard for the 109 confirmed empyemas included in this evaluation is presented in Table 3. The CAPI-PCR was positive for one or more targets in 107 samples, giving a sensitivity for the diagnosis of CAPI of 98.1% in our material. The PCR was negative for two monobacterial empyemas caused by unusual bacteria not included in the panel (one *Bacillus cereus* and one *Listeria monocytogenes*).

For on-target bacteria, the CAPI-PCR reproduced all 171 findings made by TNGS, and there was a complete concordance between the two methods for 98 (89.9%) samples. From 11 samples, the CAPI-PCR made an additional 13 identifications of which all were either confirmed by culture or an alternative PCR (10 detections) or supported by a finding below cutoff in the TNGS analysis (three detections) (Table S3).

In 69 (63.3%) samples, the PCR detected all present pathogens. These were 57 monobacterial samples and 12 polymicrobial samples (Table S4). In the remaining 38 empyemas, 16S TNGS detected an additional 1 to 31 (average 5.7; mean 3) species not included in the PCR panel per sample. These represented a range of anaerobic bacteria in addition to occasional species from the facultative genera *Actinomyces*, *Eikenella*, and *Schaalia*. The additional detections and their frequencies are listed in Table S5. Additional detections by culture were limited to 13 from eight samples. The 11 pleural fluid samples from patients with non-infectious conditions were negative by all CAPI-PCRs.

**TABLE 3** Comparison of key performance parameters for the syndromic PCR, culture, and 16S TNGS[b]

| Parameter | | PCR | Culture | 16S TNGS |
|---|---|---|---|---|
| Confirmation of bacterial presence in pleural fluid | n | 107/109 | 54/109 | 109/109 |
| | % (95% CI) | 98% (94%–100%) | 50% (40%–59%) | 100% (97%–100%) |
| Detection of all bacteria in sample | n | 69/109 | 31/109 | 98/109 (89.9%) |
| | % (95% CI) | 63% (54%–72%) | 28% (20%–38%) | 90% (83%–95%) |
| Detection of bacteria targeted by PCR | n | 184/185 | 40/185 | 171/185 |
| | % (95% CI) | 99% (97%–100%) | 22% (20%–38%) | 92% (88%–96%) |
| Detection of bacteria not targeted by PCR | n | 0/236[a] | 13/236 | 236/236 |
| | % (95% CI) | 0% (0%–2%) | 6% (3%–9%) | 100% (98%–100%) |

[a]Inferring a specificity for the CAPI-PCR of 100% (95% CI 98%–100%).
[b]n, number of samples/bacteria; CI, confidence interval.

A detailed description of the individual diagnostic sensitivities and specificities for the PCRs included in the panel and for the panel as a whole measured against the composite reference standard is provided in Table S6. The 109 empyemas contained a total of 185 PCR on-target bacteria (i.e., bacteria included in the PCR panel) and 236 off-target bacteria (i.e., bacteria not included in the PCR panel) representing 103 species from 54 bacterial genera. On this large collection of complex and relevant samples, the PCR panel obtained an on-target sensitivity of 99.5% (184/185) and a specificity of 100% (Table 3; Table S6). The single missed detection was a *Klebsiella pneumoniae* cultured from a complex polymicrobial sample that was also not detected by 16S TNGS.

## Real-life experience with the CAPI-PCR

During the first 4 months following the implementation of the fully validated CAPI-PCR in routine diagnostics at HUH, we analyzed pleural fluid samples from 100 adult patients. For 31 patients, presence of bacteria was confirmed by either culture, direct 16S rRNA Sanger sequencing, or the CAPI-PCR or combinations of these (Table 4). Using the combined results from all methods as the gold standard, the CAPI-PCR had a sensitivity of 100% for the diagnosis of CAPI. In comparison, direct 16S Sanger sequencing was positive for 12 (38.7%) samples according to our standard criteria (18) and culture for only two (6.5%). All CAPI-PCR-positive/16S rRNA PCR-negative samples had Ct-values >28 in the CAPI-PCR.

## DISCUSSION

In this study, we demonstrate that a syndromic PCR panel for the most important bacteria in CAPIs represents an effective diagnostic approach for this condition. To our knowledge, this is the first evaluation of a PCR panel designed for the diagnosis of potentially highly complex infections.

On a well-described collection of 109 pleural fluid samples, the PCR panel performed clearly better than culture. It confirmed the presence of bacteria in more samples (98.1% vs 49.5%), identified all present species in more samples (63.3% vs 28.4%), and obtained a much higher sensitivity for PCR on-target bacteria (99.5% vs 26.4%). The PCR panel also performed well when measured against 16S TNGS, although as expected TNGS detected all present bacteria in more samples (89.8% vs 63.3%). Additional identifications by TNGS represented the predicted range of oral bacteria. Importantly, we did not encounter species with potentially problematic susceptibility profiles like e.g., *Eggerthella lenta* (19) or *Bacteroides fragilis* (20, 21), that might have affected our treatment recommendations. On the other hand, TNGS failed to detect low concentrations of *S. aureus* in two samples. It also did not detect a culture-positive/PCR-negative *K. pneumoniae* in one sample.

On the collection of 109 samples, 16S TNGS obtained a sensitivity for the diagnosis of CAPI (i.e., the presence of bacteria in the pleural fluid) of 100%. However, this was

**TABLE 4** Positive results among the first 100 adult patients investigated for possible CAPI after implementation of the CAPI-PCR

| ID | Pop. | CAPI-PCR[a] | | | | | | | | | | | Cult. | MS2 |
|---|---|---|---|---|---|---|---|---|---|---|---|---|---|---|
| | | SI | FU | PM | AA | SPN | SA | SPY | HI | EB | PA | 16S | | |
| 1 | AdO | P(24) | – | P(22) | P(31) | – | – | – | – | – | – | P | P | P |
| 2 | AdO | P(30) | – | – | – | – | – | – | – | – | – | – | P | P |
| 3 | AdO | – | P(32) | P(36) | – | – | – | – | – | – | – | – | – | P |
| 4 | AdO | P(21) | – | – | – | – | – | – | – | – | – | P | – | P |
| 5 | AdO | P(34) | – | – | – | – | – | – | – | – | – | – | – | P |
| 6 | AdO | P(37) | – | – | – | – | – | – | – | – | – | – | – | P |
| 7 | AdO | P(21) | P(20) | P(21) | – | – | – | – | – | – | – | P | – | P |
| 8 | AdO | P(17) | P(16) | P(16) | – | – | – | – | – | – | – | P | P | P |
| 9 | AdO | – | P(36) | – | – | – | – | – | – | – | – | – | – | P |
| 10 | AdO | P(40) | – | – | – | – | – | – | – | – | – | – | – | P |
| 11 | AdO | P(31) | – | – | – | – | – | – | – | – | – | – | – | P |
| 12 | AdO | – | P(38) | – | – | – | – | – | – | – | – | P | – | P |
| 13 | AdO | P(28) | – | – | – | – | – | – | – | – | – | – | – | P |
| 14 | AdO | P(30) | – | – | – | – | – | – | – | – | – | – | – | P |
| 15 | AdO | P(32) | P(26) | P(27) | – | – | – | – | – | – | – | P | – | P |
| 16 | AdO | – | P(24) | – | – | – | – | – | – | – | – | P | – | P |
| 17 | AdO | P(23) | – | – | – | – | – | – | – | – | – | P | – | P |
| 18 | AdS | – | – | – | – | – | – | – | P(38) | – | – | – | – | P |
| 19 | AdS | – | – | – | – | – | – | P(29) | – | – | – | – | – | P |
| 20 | AdS | – | – | – | – | – | – | – | P(35) | – | – | – | – | P |
| 21 | AdS | – | – | – | – | P(28) | – | – | – | – | – | P | – | P |
| 22 | AdS | – | – | – | – | P(37) | – | – | – | – | – | – | – | P |
| 23 | AdS | – | – | – | – | P(26) | – | – | – | – | – | P | – | P |
| 24 | AdS | – | – | – | – | – | P(38) | – | – | – | – | – | – | P |
| 25 | AdS | – | – | – | – | P(24) | – | – | – | – | – | P | – | P |
| 26 | AdS | – | – | – | – | – | – | – | P(37) | – | – | – | – | P |
| 27 | AdS | – | – | – | – | – | P(37) | – | – | – | – | – | – | P |
| 28 | AdS | – | – | – | – | P(23) | – | – | – | – | – | P | – | P |
| 29 | AdS | – | – | – | – | P(37) | – | – | – | – | – | – | – | P |
| 30 | AdS | – | – | – | – | P(35) | – | – | – | – | – | – | – | P |
| 31 | AdS | – | – | – | – | – | P(39) | – | – | – | – | – | – | P |

[a]ID, sample identity; Pop, population; SI, *S. intermedius/constellatus*-PCR; FU, *Fusobacterium*-PCR; PM, *P. micra*-PCR; AA, *A. aphrophilus*-PCR; SPN, *S. pneumoniae*-PCR; SA, *S. aureus*-PCR; SPY, *S. pyogenes*-PCR; HI, *H. influenzae*-PCR; EB, *Enterobacteriales*-PCR; PA, *P. aeruginosa*-PCR; 16S, 16S rRNA-PCR; Cult., culture; AdO, adult (≥18 years) oral-type pleural infection; AdS, adult (≥18 years) pleural infection secondary to CAP; P, positive; – = negative/not detected; a, PCR Ct-values in parenthesis.

attributed to a positive culture and/or a positive 16S rRNA PCR being one of the inclusion criteria, and such high sensitivity is unlikely to be achievable in routine clinical practice. Even in labs that have implemented 16S TNGS as a diagnostic service, a universal 16S rRNA real-time PCR is still being used as a rapid and low-cost screening method to filter out putative negative specimens prior to the labor intensive and costly sequencing step (22). Although criteria for a positive sample can be less stringent than those commonly used for direct 16S Sanger sequencing (22), this will inevitably lead to the exclusion of some weak positive samples. Therefore, in a routine clinical setting, a PCR panel is likely to obtain a higher sensitivity than 16S TNGS, as indicated by the data presented in Table 4. Alternatively, the CAPI-PCR can be used as a better approach to rule in empyema eligible for a full description by 16S TNGS. In such an approach, the PCR results can be reported directly as a rapid and preliminary guidance. However, as demonstrated in this study, the additional clinical value of performing a complete microbial description becoming available several days later is likely to be limited.

It must be clearly communicated to the doctors in charge of the patient that this is a pragmatic PCR panel targeting the most common causes of CAPI, that additional off-target species can be present and that a negative PCR does not rule out the diagnosis of a bacterial pleural infection. For patients with a strong clinical suspicion of CAPI and a negative CAPI-PCR, we still recommend the use of 16S rRNA-based diagnostic approaches. Clinicians must also understand that although *F. nucleatum* and *A. aphrophilus* can both cause monobacterial infections, they should always be interpreted as indicators of a more complex predominantly anaerobic infection. At HUH, we solve this by using standard comments on the reports as described in Table 2. We emphasize that this approach has been developed and evaluated for CAPIs only. In postoperative pleural infections, pleural infections after rupture of the esophagus or related to metastatic cancer, broader and more unpredictable spectrums of microbes can be involved.

The PCR panel is run daily in our laboratory at HUH and has significantly increased sensitivity and shortened time to actionable results for CAPIs. Ideally, the panel should be available on an automated PCR system, reducing hands-on time and permitting immediate analysis of a sample after arrival to the lab. Several automated PCR systems allow for a larger number of targets than included in the presented panel. Useful additions would include species-specific PCRs for a few of the most common *Enterobacteriaceae* like *K. pneumoniae* and *Escherichia coli,* and in many regions, a PCR for *Mycobacterium tuberculosis* is relevant. Selected markers for antimicrobial resistance would also be valuable.

We see this article as a proof of concept and believe that syndromic PCR panels can be of value also in other complex polymicrobial infections, provided key microbes for identification and treatment can be defined. It is well established that culture-based diagnostics are insufficient for a range of invasive purulent infections such as brain abscesses (23) and intra-abdominal infections (24–26). Our group recently confirmed a strong resemblance between the microbial flora of OPI and that of oral-sinus-derived brain abscesses (6). We suggest that the panel described for CAPI in this paper, with only a few adjustments, could be developed into a syndromic panel for community-acquired brain abscesses, eventually a joint panel for both conditions. Relevant modifications could be the inclusion of PCRs for *Nocardia* spp. and *Cryptococcus neoformans*

The strengths of our study include the large and well-characterized collection of clinical samples used in the validation of the CAPI-PCR concept and the comparison of the CAPI-PCR to both culture and 16S TNGS. The data from routine clinical use of the CAPI-PCR were provided to indicate the potential gain in sensitivity as compared to current diagnostic approaches. For these routine samples, we did not attempt to confirm findings made exclusively by PCR using another method, since the CAPI-PCR had already demonstrated an excellent specificity in the validation and since the positive PCRs correlated well with the clinical conditions of the patients. Future research should include a more systematic evaluation of the CAPI-PCR in a clinical setting, including real-life

diagnostic sensitivity and specificity, positive and negative predictive values, turnaround times, and consequences for patient treatment. The low sensitivity for culture is believed to be mainly due to antimicrobial treatment prior to sample collection. However, it should also be mentioned that most samples in our routine diagnostics were received on sterile containers only. Bedside inoculation of pleural fluid into blood culture bottles as a supplement has been shown to increase culture positivity rates with 50% (27, 28). This would not have affected our conclusions since culture obtained a sensitivity of only 6.5% as compared to PCR in this setting.

We conclude that a syndromic PCR panel for CAPIs represents a rapid and sensitive alternative to current diagnostic approaches. It can dramatically improve sensitivity in laboratories that today depend upon culture-based diagnostics. Data from routine clinical practice indicate that it might also perform better than approaches based on amplification and sequencing of the bacterial 16S rRNA gene.

## ACKNOWLEDGMENTS

The Department of Microbiology, Haukeland University Hospital, funded the PCR and 16S rRNA NGS analyses.

Ø.K. and R.D. conceived and designed the study and prepared the ethics protocol. T.M.E., Ø.F., W.S., B.B., F.G., A.K.-A., and R.T. enrolled patients and collected samples and clinical data. T.M.L., H.F., K.W.L., F.G., and R.H. were responsible for the study at the local hospitals, and responsible for the local microbiological diagnostic procedures and sending of samples to HUH for 16S TNGS and CAPI-PCR. Ø.K. designed the CAPI-PCR primers and probes. S.K.Ø.H. and M.T. validated the PCRs, implemented them in the routine diagnostics, and conducted the PCR experiments. M.T. performed the 16S TNGS experiments. R.D. and Ø.K. analyzed the 16S TNGS data. Ø.K. and R.D. wrote the first version of the manuscript. Ø.K. and R.D. supervised the study. All authors revised and approved the final version of the manuscript. All authors had full access to all the data in the study and had final responsibility for the decision to submit for publication. Ø.K. and R.D. verified the underlying data of the study.

## AUTHOR AFFILIATIONS

[1]Department of Microbiology, Haukeland University Hospital, Bergen, Norway

[2]Department of Clinical Science, University of Bergen, Bergen, Norway

[3]Department of Thoracic Medicine, Haukeland University Hospital, Bergen, Norway

[4]Division of Medicine and Laboratory Sciences, Institute of Clinical Medicine, Faculty of Medicine, University of Oslo, Oslo, Norway

[5]Department of Microbiology and Infection Control, Akershus University Hospital, Lørenskog, Akershus, Norway

[6]Department of Pulmonary Medicine, Akershus University Hospital, Lorenskog, Akershus, Norway

[7]Department of Microbiology, Stavanger University Hospital, Stavanger, Norway

[8]Department of Pulmonary Medicine, Stavanger University Hospital, Stavanger, Norway

[9]Department of Medicine, Haraldsplass Deaconess Hospital, Bergen, Norway

[10]Department of Medical Microbiology, St. Olavs Hospital, Trondheim University Hospital, Trondheim, Norway

[11]Department of Thoracic Medicine, St. Olavs Hospital, Trondheim University Hospital, Trondheim, Norway

[12]Department of Microbiology, Førde Central Hospital, Førde, Norway

[13]Department of Medicine, Førde Central Hospital, Førde, Norway

## AUTHOR ORCIDs

Øyvind Kommedal http://orcid.org/0000-0002-4390-0370

Ruben Dyrhovden http://orcid.org/0000-0002-2486-5426

## AUTHOR CONTRIBUTIONS

Øyvind Kommedal, Conceptualization, Data curation, Formal analysis, Investigation, Methodology, Project administration, Supervision, Writing – original draft | Tomas Mikal Eagan, Resources, Writing – review and editing | Øystein Fløtten, Resources, Writing – review and editing | Truls Michael Leegaard, Resources, Writing – review and editing | William Siljan, Resources, Writing – review and editing | Hilde Fardal, Resources, Writing – review and editing | Bjørnar Bø, Resources, Writing – review and editing | Fredrik Grøvan, Resources, Writing – review and editing | Kjersti Wik Larssen, Resources, Writing – review and editing | Arne Kildahl-Andersen, Resources, Writing – review and editing | Reidar Hjetland, Resources, Writing – review and editing | Rune Tilseth, Resources, Writing – review and editing | Sølvi Kristine Øyen Hareide, Investigation, Validation, Writing – review and editing | Marit Tellevik, Investigation, Validation, Writing – review and editing | Ruben Dyrhovden, Conceptualization, Data curation, Formal analysis, Investigation, Methodology, Project administration, Supervision, Writing – review and editing

## DATA AVAILABILITY

The 16S rRNA NGS data have been deposited in the European Nucleotide Archive (ENA) at EMBL-EBI under accession number PRJEB62005. Other source data of this study are available from the corresponding author upon request.

## ETHICS APPROVAL

The study was approved by the regional ethical committee of South-East Norway (REK 31938). Written informed consent was obtained from all participants.

## ADDITIONAL FILES

The following material is available online.

### Supplemental Material

**Document S1 (Spectrum03510-23-s0001.docx).** PCR target details.
**Supplemental Tables (Spectrum03510-23-s0002.docx).** Tables S1 to S6.

### Open Peer Review

**PEER REVIEW HISTORY (review-history.pdf).** An accounting of the reviewer comments and feedback.

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
