## [Reviewer comments · Microbiology Spectrum]

Microbiology Spectrum

Microbiological diagnosis of pleural infections. A comparative evaluation of a novel syndromic real-time PCR panel

Øyvind Kommedal, Eagan Tomas, Øystein Fløtten, Truls Leegaard, William Siljan, Hilde Fardal, Bjørnar Bø, Fredrik Grøvan, Kjersti Larssen, Arne Kildahl-Andersen, Reidar Hjetland, Rune Tilseth, Sølvi Hareide, Marit Tellevik, and Ruben Dyrhovden

Corresponding Author(s): Øyvind Kommedal, Haukeland Universitetssjukehus

Review Timeline:

Submission Date:	September 29, 2023
Editorial Decision:	January 2, 2024
Revision Received:	February 8, 2024
Accepted:	February 19, 2024

Editor: Kessendri Reddy

Reviewer(s): The reviewers have opted to remain anonymous.

Transaction Report:

DOI: <https://doi.org/10.1128/spectrum.03510-23>

Re: Spectrum03510-23 (Microbiological diagnosis of pleural infections. A comparative evaluation of a novel syndromic real-time PCR panel)

Dear Dr. Øyvind Kommedal:

Thank you for the privilege of reviewing your work. Below you will find my comments, instructions from the Spectrum editorial office, and the reviewer comments.

Revision Guidelines

Sincerely,
Kessendri Reddy
Editor
Microbiology Spectrum

Additional comments:

Please include a section in the methods on data management and statistical analysis, including sample size calculation if this was performed.

It is recommended that 95% confidence intervals be generated for the performance of the assays reviewed to allow the reader context on the accuracy of these estimates.

Reviewer #1 (Comments for the Author):

The paper of Kommedal et al. focuses on an important topic, the community acquired pleural infections. These are characterized by high mortality and morbidity. Pleural infections are often polymicrobial and this associated to the problem of the low sensitivity of pleural fluid cultures and to the fact that the bacteriological etiology of pleural infection is quite different from that of pneumonia make extremely important to identify the bacteria involved in these infections. The authors developed a PCR panel comprising four duplex and one triplex PCRs able to identify 11 bacteria commonly implicated in this pathology. However, I have some concerns that need to be addressed:

Major points:

Material and Methods

PCR design and target considerations

The Authors claim: "To allow for a single set up, all PCRs were adjusted to the same thermal profile". In which way? How they can confirm that there is not a lack of sensitivity for some of the included pathogens with the use of the same thermal profile? An inhibition control was included in the panel. What about an extraction control?

No culture conditions were reported

Results

The method was applied to the analysis of pleural fluids from 100 patients. The results show that 31 were positive with CAPI-PCR. The comparison was made with 16S rRNA Sanger sequencing and culture. Why was not used the same assay used previously (TNGS)? What about the other samples? Were all negative? It was important to have the results of TNGS also on these samples.

Then in the Table was reported only ten and not 11 pathogens. Why? And the results obtained with the internal control should also be included

What about the antibiotic profile of all these bacteria (for those isolated)? What was the clinical outcome of the patients for which was given a molecular result?

Minor point:

The table with the result of the 110 pleural fluid should be included in the principal text and not in the supplemental materials.

Reviewer #2 (Comments for the Author):

There are a few suggestions, authors need to rework.

1. Robust methodology for CAPI PCR is lacking. In detail, authors may shed light on analytical sensitivity, specificity of their newly developed assay and also diagnostic sensitivity, specificity, PPV, NPV etc in comparison to gold standard
2. There is a mention about multicentric study, however sample size is very low, authors may substantiate it or increase the sample size to draw a fruitful conclusion
3. References years mentioned in texts and the bibliography need to cross checked

Point-by-point Response to Reviewers

We appreciate the thorough work of the reviewers, and their valuable suggestions and comments.

Additional comments:

Comment: Please include a section in the methods on data management and statistical analysis, including sample size calculation if this was performed.

Response: Section included (Lines 204-220).

Comment: It is recommended that 95% confidence intervals be generated for the performance of the assays reviewed to allow the reader context on the accuracy of these estimates.

Response: Provided for all relevant parameters in novel Supplementary Table S6. Also included in Table 3.

Reviewer #1 (Comments for the Author):

The paper of Kommedal et al. focuses on an important topic, the community acquired pleural infections. These are characterized by high mortality and morbidity. Pleural infections are often polymicrobial and this associated to the problem of the low sensitivity of pleural fluid cultures and to the fact that the bacteriological etiology of pleural infection is quite different from that of pneumonia make extremely important to identify the bacteria involved in these infections. The authors developed a PCR panel comprising four duplex and one triplex PCRs able to identify 11 bacteria commonly implicated in this pathology.

However, I have some concerns that need to be addressed:

Major points:

Material and Methods

PCR design and target considerations

Comment: The Authors claim: "To allow for a single set up, all PCRs were adjusted to the same thermal profile". In which way? How they can confirm that there is not a lack of sensitivity for some of the included pathogens with the use of the same thermal profile?

Response: Thank you for pointing this out. This was just a matter of poor wording from our side. The PCRs were designed specifically to fit the same thermal profile, and no compromise on PCR conditions were necessary. We have changed the wording accordingly (Lines 173-175).

Comment: An inhibition control was included in the panel. What about an extraction control?

Response: It is an extraction and inhibition control. We have corrected the text and included the spiking of sample material with MS2 DNA plasmid in the material and methods section. The extraction and inhibition control was implemented in relation to the transfer of the test to the diagnostic routine. The 109 Samples included in the validation had not been spiked with MS2 prior to extraction as part of the original studies (Lines 182-185 and line 558-559/Table 1).

Comment: No culture conditions were reported

Response: Culture conditions for the post-validation 100 routine samples have been included (Lines 227-234).

Results

Comment: The method was applied to the analysis of pleural fluids from 100 patients. The results show that 31 were positive with CAPI-PCR. The comparison was made with 16S rRNA Sanger sequencing and culture. Why was not used the same assay used previously (TNGS)? What about the other samples? Were all negative? It was important to have the results of TNGS also on these samples.

Response: The main purpose of this paper was to describe the principle behind the pragmatic PCR i.e. how in-depth understanding of a potentially very complex bacterial infections can allow you to detect only a limited number of key pathogens necessary for confirming the infection and guide antimicrobial treatment, and further to validate this approach on a large collection of well characterized relevant clinical samples. This was done using the 109 well-characterized samples from the two previous studies.

Following this validation, we implemented the novel PCR in our diagnostic routine at Haukeland University Hospital. At this point - the PCR concept had been completely validated (as indicated in Figure 1), and the results from the PCR could therefore be accepted as a surrogate for NGS. These data from our diagnostic routine was included only to illustrate that in a real-life diagnostic setting, the PCR might potentially be more sensitive than even 16S TMS. The reason for this is that most labs will use a broad-range real-time 16S rRNA PCR to rule in or rule out samples eligible for 16S TMS (most pleural fluids will be from non-infectious conditions like e.g. cancer or heart-failure). Due to background contamination of reagents, a 16S rRNA real-time PCR can only discriminate reliably between positive and negative samples up to around PCR cycle 28-30 as compared to a species specific PCR that can reliably detect pathogens typically up to PCR cycle 36-38. This point is not illustrated by

the validation of the 109 samples, since all these samples were included either based on a positive 16S rRNA PCR or a positive culture and therefore automatically resulted in a 100% diagnostic sensitivity for the 16S TMS assay. This is an important part of our discussion (Lines 307-321).

For the 100 routine samples, as the reviewer point out, there is a possibility that some of the samples that were negative by culture, negative by the broad-range 16S rRNA PCR and negative by our novel syndromic PCR, could still have yielded a relevant finding if investigated by TNGS. This would then need to be samples containing low concentrations of DNA from bacteria not normally involved in community-acquired pleural infections.

Although theoretically possible, it would be a rare event (in our prospective study we found only two atypical infections, one caused by *Bacillus cereus* and one by *Listeria monocytogenes*, both positive by the 16S rRNA PCR), not likely to affect the results presented in Table 4 significantly. To find such eventual infections, we would need to analyze all samples using TNGS regardless of the result from the 16S rRNA PCR.

Nevertheless, the results from the first 100 patient samples must be considered a preliminary indication of the real-life performance of our PCR. We fully agree that to establish the real-life diagnostic sensitivity of our PCR, a prospective evaluation in a diagnostic setting should be conducted and that such evaluation should also include a head-to-head comparison with 16S TNGS. This will require a lot more than 100 samples. We are in the process of setting up such a study together with a major US hospital. However, our approach is novel, and to gain acceptance, the description of the principle and the evaluation of the diagnostic sensitivity on a well described material as provided in the present manuscript must be the first steps. We emphasize in the discussion that: “Future research should include a more systematic evaluation of the CAPI-PCR in a clinical setting, including real-life diagnostic sensitivity and

specificity, positive and negative predictive values, turn-around times and consequences for patient treatment” (Lines 359-362).

Comment: Then in the Table was reported only ten and not 11 pathogens. Why? And the results obtained with the internal control should also be included

Response: There are two PCRs covering the Fusobacteria. One is targeting the *F. nucleatum* complex and the other *F. necrophorum/gonidiaformans*. In our assay we have chosen to use the same fluorophore dye on both probes, so the results cannot be distinguished. They are reported as “Fusobacterium sp.” It is mentioned in the footnotes of Table 1 that “d = optionally two separate fluorophores can be used to distinguish between F_{necg} and F_{nucl}”. The results from the positive control have been included in Table 4 as requested.

Comment: What about the antibiotic profile of all these bacteria (for those isolated)? What was the clinical outcome of the patients for which was given a molecular result?

Response: It is a weakness for PCR-based detection that a susceptibility profile cannot be provided. We have not assessed specifically the antibiotic profiles for the cultured isolates in this study. Fortunately, as we discuss in the manuscript, bacteria involved in OPI have relatively unproblematic antimicrobial susceptibility profiles, allowing us to guide antimicrobial treatment based on the detections as shown in the standard comments suggested in Table 2.

In the 2023 prospective study, we found an in-hospital mortality of 11% and a 1-year mortality of 17% for patients with oral-type infections. Mortality was confined to patients with advanced age and/or significant comorbidities. We do not have any outcome data on the patients presented in Table 4.

We already point out in the discussion that “Future research should include a more systematic evaluation of the CAPI-PCR in a clinical setting, including ... and consequences for patient treatment.” (Lines 359-362).

Minor point:

Comment: The table with the result of the 110 pleural fluid should be included in the principal text and not in the supplemental materials.

Response: Thank you. Now included as Table 4.

Reviewer #2 (Comments for the Author):

There are a few suggestions, authors need to rework.

Comment: Robust methodology for CAPI PCR is lacking. In detail, authors may shed light on analytical sensitivity, specificity of their newly developed assay and also diagnostic sensitivity, specificity, PPV, NPV etc in comparison to gold standard

Response: We have included a new Supplementary Table S6 providing the individual diagnostic sensitivities and specificities for all the PCRs included in the panel individually and for the PCR panel as a whole as compared to the composite reference standard.

However, as we now state in the discussion, the real-life diagnostic sensitivity and specificity of the test including PPV and NPVs will need to be established in a future study where the PCR is implemented as a standard primary test (Lines 359-362).

Comment: There is a mention about multicentric study, however sample size is very low, authors may substantiate it or increase the sample size to draw a fruitful conclusion

Response: The validation of the PCR is based on 109 samples, 36 samples from a single center retrospective study and 76 samples from later prospective multicenter study (Figure 1). This is a PCR intended specifically for community-acquired pleural infections, excluding e.g. cancer related empyema and post-surgery empyema. Community-acquired pleural infection is a relatively rare condition. The incidence of non-cancer, non-surgery pleural empyema was estimated to 4.7/100.000 in a recent French study which was in line with our calculations based on the single center retrospective study. The hospitals participating in the prospective study had a total catchment area of 1.5 million which theoretically should have given around 120-150 cases over two years. However, as explained in reference 6 (Dyrhovden et al., 2023), the COVID-19 pandemic severely affected recruitment in three hospitals resulting in the final inclusion of 77 patients.

We respectfully disagree that 109 is a very low sample size in this context. In our opinion, the inclusion of 109 positive clinically well characterized samples in a PCR validation is sufficient for a robust evaluation of the diagnostic performance of the test on a relevant sample material. A recent evaluation of a commercial PCR panel developed for pneumonia in the diagnosis of patients with pleural effusion (Franchetti et al., reference 13 in our manuscript) included only 61 confirmed empyema (both community and hospital acquired). Notably they obtained an overall low sensitivity and concluded that “A dedicated pleural empyema multiplex PCR panel including anaerobes would be needed to cover most common pathogens involved in pleural infection.” It should also be mentioned that their study did not include complete microbial characterizations by 16 TMS.

Comment: These are only the community-acquired infections. Hospital-acquired infections were excluded.

Response: The underlying principle for this pragmatic PCR approach is the characteristic microbial patterns described for oral-type community acquired empyema. Such consistent patterns cannot be expected in hospital-acquired infections often associated with surgery, rupture of the esophagus, trauma or cancer. For these infections there is often disruption of natural physiological and physical barriers against bacterial invasion and much more complex, unpredictable and even dynamic microbial compositions must be expected. This is addressed in Lines 330-333.

Comment: References years mentioned in texts and the bibliography need to cross checked

Response: Thank you. Corrected from 2018 to 2019 for Dyrhovden et al. in the Figure 1 legend.

Re: Spectrum03510-23R1 (Microbiological diagnosis of pleural infections. A comparative evaluation of a novel syndromic real-time PCR panel)

Dear Dr. Øyvind Kommedal:

Your manuscript has been accepted, and I am forwarding it to the ASM production staff for publication. Your paper will first be checked to make sure all elements meet the technical requirements. ASM staff will contact you if anything needs to be revised before copyediting and production can begin. Otherwise, you will be notified when your proofs are ready to be viewed.

Sincerely,
Kessendri Reddy
Editor
Microbiology Spectrum